# Functionally important residues from graph analysis of coevolved dynamic couplings

**Manming Xu[1†], Sarath Chandra Dantu[2†], James A Garnett[3], Robert A Bonomo[4,5,6,7,8], Alessandro Pandini[2*], Shozeb Haider[1,9,10*†]**

[1]UCL School of Pharmacy, London, United Kingdom; [2]Department of Computer Science, Brunel University London, Uxbridge, United Kingdom; [3]Centre for Host-Microbiome Interactions, Faculty of Dentistry, Oral & Craniofacial Sciences, King's College London, London, United Kingdom; [4]Research Service, Louis Stokes Cleveland Department of Veterans Affairs Medical Center, Cleveland, United States; [5]Department of Molecular Biology and Microbiology, Case Western Reserve University School of Medicine, Cleveland, United States; [6]Department of Medicine, Case Western Reserve University School of Medicine, Cleveland, United States; [7]Departments of Pharmacology, Biochemistry, and Proteomics and Bioinformatics Case Western Reserve University School of Medicine, Cleveland, United States; [8]CWRU-Cleveland VAMC Center for Antimicrobial Resistance and Epidemiology (Case VA CARES), Cleveland, United States; [9]University of Tabuk (PFSCBR), Tabuk, Saudi Arabia; [10]UCL Center for Advanced Research Computing, University College London, London, United Kingdom

**\*For correspondence:**
alessandro.pandini@brunel.ac.uk (AP);
shozeb.haider@ucl.ac.uk (SH)

[†]These authors contributed equally to this work

## eLife Assessment

This article reports the analysis of coevolutionary patterns and dynamical information for identifying functionally relevant sites. These findings are considered **important** due to the broad utility of the unified framework and network analysis capable of revealing communities of key residues that go beyond the residue-pair concept. The data are **solid** and the results are clearly presented.

**Abstract** The relationship between protein dynamics and function is essential for understanding biological processes and developing effective therapeutics. Functional sites within proteins are critical for activities such as substrate binding, catalysis, and structural changes. Existing computational methods for the predictions of functional residues are trained on sequence, structural, and experimental data, but they do not explicitly model the influence of evolution on protein dynamics. This overlooked contribution is essential as it is known that evolution can fine-tune protein dynamics through compensatory mutations either to improve the proteins' performance or diversify its function while maintaining the same structural scaffold. To model this critical contribution, we introduce DyNoPy, a computational method that combines residue coevolution analysis with molecular dynamics simulations, revealing hidden correlations between functional sites. DyNoPy constructs a graph model of residue–residue interactions, identifies communities of key residue groups, and annotates critical sites based on their roles. By leveraging the concept of coevolved dynamical couplings—residue pairs with critical dynamical interactions that have been preserved during evolution—DyNoPy offers a powerful method for predicting and analysing protein evolution and dynamics. We demonstrate the effectiveness of DyNoPy on SHV-1 and PDC-3, chromosomally encoded β-lactamases linked to antibiotic

resistance, highlighting its potential to inform drug design and address pressing healthcare challenges.

## Introduction

Quantifying the contribution of individual residues or residue groups to protein function is important to estimate the pathogenic effect of mutations (*Stenson et al., 2017*). Identifying the functional roles of individual residues has primarily been done through mutagenesis experiments (*Matreyek et al., 2018*). Bioinformatics methods have complemented these approaches through analysis of multiple sequence alignments (MSA) of homologous proteins and structural data (*Poelwijk et al., 2016*; *Høie et al., 2022*; *Blaabjerg et al., 2023*; *Dunham and Beltrao, 2021*; *Hopf et al., 2017*; *Radivojac et al., 2013*). Among these methods, computational techniques that can decode inter-residue evolutionary relationships from MSAs have paved the way for machine learning (ML)-based strategies that can predict protein structure (*Jumper et al., 2021*; *Lin et al., 2023*; *Baek et al., 2021*; *Marks et al., 2012*), stability (*Broom et al., 2020*), and function (*Hopf et al., 2017*) and extend the scope of computational protein design (*Ding et al., 2024*; *Wu et al., 2019*; *Russ et al., 2020*). A most recent approach has combined experimental data from three proteins, NUDT15, PTEN, and CYP2C9, on the stability and function with sequence and structural features to train an ML model to predict functional sites (*Cagiada et al., 2023*).

Functional sites are often regulated by both local and global interactions. Changes in these interactions are instrumental for functional events like substrate binding, catalysis, and conformational changes (*Wodak et al., 2019*). The development of physical models of protein dynamics and the increase in available computational power has stimulated the adoption of computational techniques (*Campitelli et al., 2020*; *Rodrigues et al., 2021*) to investigate the conformational dynamics of proteins, an essential component of the many biological functions (*Henzler-Wildman and Kern, 2007*; *James and Tawfik, 2003*). Different models have been proposed to describe the interactions between residues during simulations and network models have been particularly popular, including methods on single structures and molecular dynamics (MD) simulations data built by analysing the response to external forces on residue networks (*Nevin Gerek et al., 2013*), estimating the prevalence of non-covalent energy interaction networks in homologous proteins (*Yehorova et al., 2024*), or analysing linear or non-linear correlation in atomic fluctuations (*Lange and Grubmüller, 2006*; *Osuna, 2021*). These techniques have demonstrated their usefulness in extracting allosteric networks from structural data with applications in enzyme design (*Osuna, 2021*).

However, none of these techniques incorporate information on residue evolution into the computational approach, while it has been established that evolution through compensatory mutations in dynamic regions, like hinges and loops, can fine-tune protein structural dynamics and introduce promiscuity, thereby diversifying biological function. Assuming that protein functional dynamics is conserved during evolution, significant information on dynamic regions and substrate recognition sites should be recoverable using inter-residue coevolution scores extracted from MSAs (*Granata et al., 2017*; *Liu and Bahar, 2012*). Coevolution analysis and MD simulations have independently (*Parente et al., 2015*) and synergistically been combined in the past to identify important residues for function (*Ponzoni et al., 2015*; *Sutto et al., 2015*; *Estabrook et al., 2005*; *Chen et al., 2011*; *Wang et al., 2013*). Yet a method that combines hidden information on dynamics from evolution with direct information on local and global dynamics from conformational ensembles from MD is not yet available.

Here, we present DyNoPy, a computational method that can extract hidden information on functional sites from the combination of pairwise residue coevolution data and powerful descriptors of dynamics extracted from the analysis of MD ensembles. The method can detect coevolved dynamic couplings, that is, residue pairs with critical dynamical interactions that have been preserved during evolution. These pairs are extracted from a graph model of residue–residue interactions. Communities of important residue groups are detected, and critical sites are identified by their eigenvector centrality in the graph (*Figure 1*). We demonstrate the power of this approach on SHV-1 and PDC-3 β-lactamases of major clinical importance (*Olehnovics et al., 2021*; *Chen et al., 2024*). DyNoPy successfully detects residue couplings that align with previous studies, guide in the explanations of mutation sites

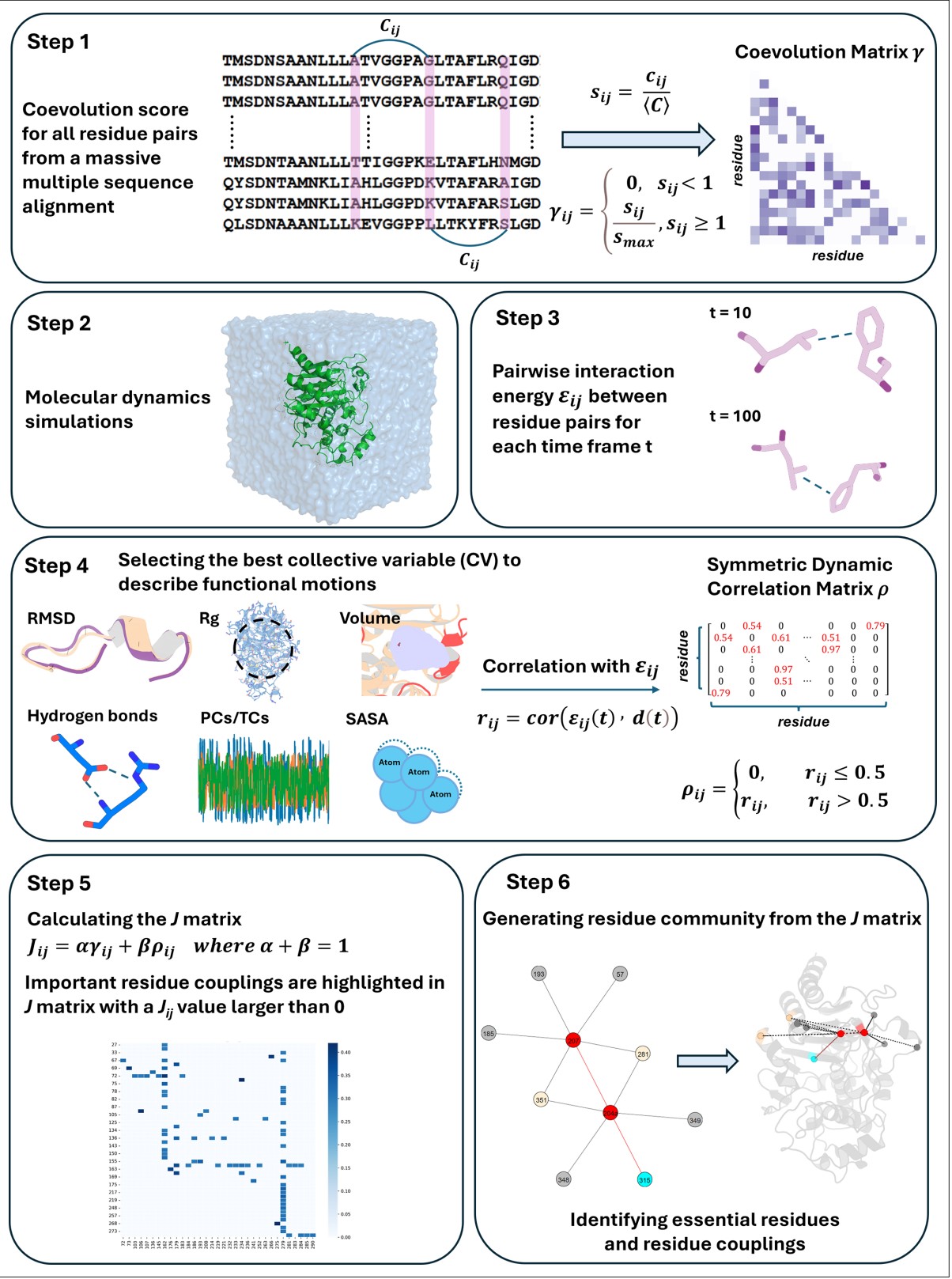

**Figure 1.** Overview of the DyNoPy workflow.

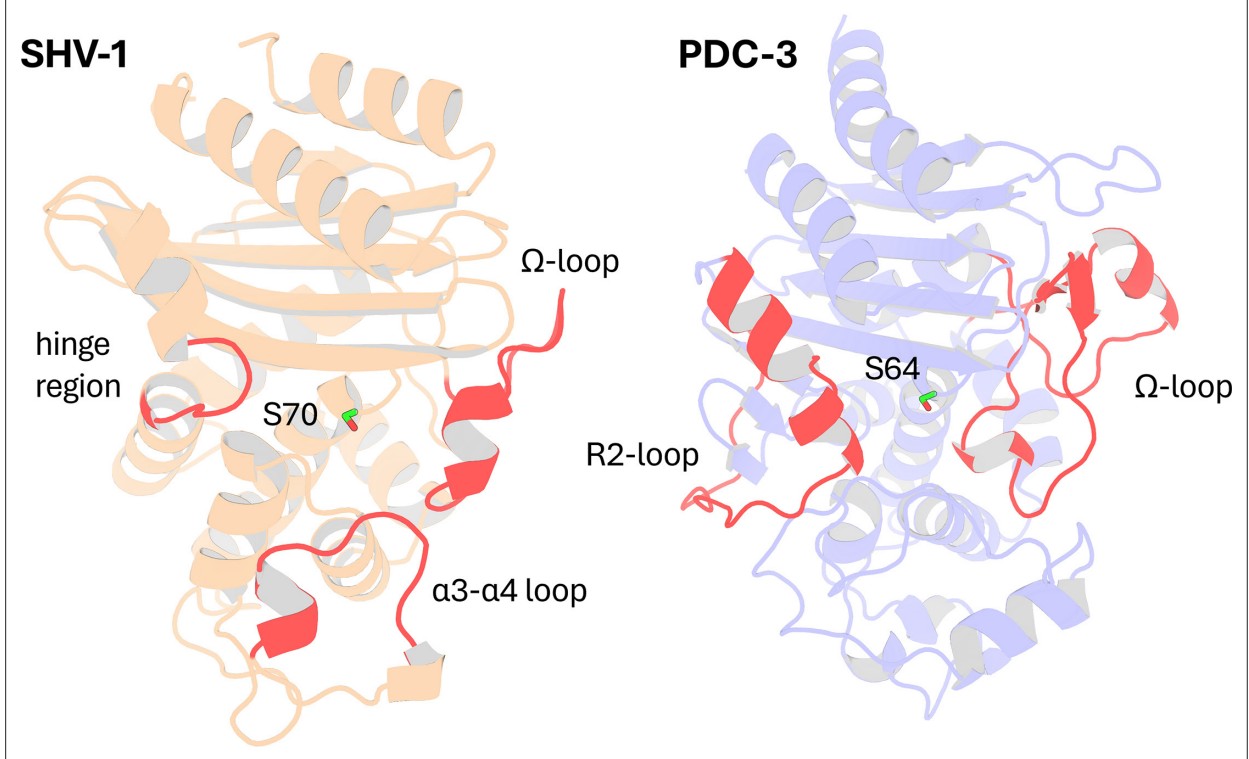

**Figure 2.** Structural Comparison of SHV-1 (PDB ID: 3N4I) and PDC-3 (PDB ID: 4HEF) β-lactamases. Catalytic serine $S_{70}$ (SHV-1) and $S_{64}$ (PDC3) are highlighted using stick representation. Important loops surrounding the active site are highlighted in red. In SHV-1, highlighted loops are the α3-α4 loop (residues 101–111), the Ω-loop (residues 164–179), and the hinge region (residues 213–218). In PDC-3, highlighted loops are the Ω-loop (residues 183–226) and the R2-loop (residues 280–310).

The online version of this article includes the following figure supplement(s) for figure 2:

**Figure supplement 1.** SHV-1 structural nomenclature.

**Figure supplement 2.** PDC-3 structural nomenclature.

**Figure supplement 3.** Crystal structure and binding pocket of SHV-1 (PDB ID: 3N4I) and PDC-3 (PDB ID: 4HEF).

with previously unexplained mechanisms, and provide predictions on plausible important sites for the emergence of clinically relevant variants.

## Results and discussion

β-Lactamases are a group of enzymes capable of hydrolysing β-lactams, conferring resistance to β-lactam antibiotics (*Poole, 2004*). These enzymes are evolving rapidly as single amino acid substitutions are sufficient to drive their evolution and increase their catalytic spectrum and inhibitor resistance profile (*Bush, 2018*). The widespread dissemination of β-lactamases across different bacterial species and their extensive emergence highlight their global impact on antibiotic resistance (*Bush, 2013*). The rapid evolution of β-lactamases and their clinical significance (*Bush, 2018*) makes them an ideal target for evaluating the robustness of DyNoPy.

In this study, we applied DyNoPy to two model enzymes from different β-lactamase families: class A β-lactamase SHV-1 (a chromosomally encoded enzyme in *Klebsiella pneumoniae*) and class C β-lactamase PDC-3 (a chromosomally encoded enzyme in *Pseudomonas aeruginosa*) (*Olehnovics et al., 2021*; *Chen et al., 2024*; *Figure 2*). Both class A and class C β-lactamases comprise an α/β domain and an α helical domain, with the active site situated in between (*Matagne et al., 1998*; *Philippon et al., 2022*). Moreover, both enzymes target the carbonyl carbon of the β-lactams using a highly conserved serine residue (*Palzkill, 2018*; *Jacoby, 2009*). Despite these similarities, the structures of class A and class C β-lactamases are remarkably different (*Figure 2—figure supplements 1 and 2*). In class A β-lactamases, the active site is surrounded by three loops: the α3-α4 loop (residues 101–111),

the Ω-loop (residues 164–179), and the hinge region (residues 213–218) (*Galdadas et al., 2021*). The Ω-loop is particularly critical as it positions $N_{170}$ to hydrogen bond with the $E_{166}$ via a conserved water molecule, which is essential for initiating the deacylation step (*Kuzin et al., 1999*). Compared to class A β-lactamases, the active site of class C β-lactamases is wider, conferring a broader substrate binding capability (*Medeiros, 1997*; *Figure 2—figure supplement 3*). The active site of class C β-lactamases can be divided into two parts: the R1 site and the R2 site (*Jacoby, 2009*). The R1 region is surrounded by the extended Ω-loop (residues 183–226), while the R2 site is enclosed by the R2-loop (residues 280–310) (*Chen et al., 2024*). The Ω-loop in class C β-lactamases is significantly longer than that in class A, enhancing the active site ability to accommodate diverse substrates and contributing to the extended spectrum profile of some class C enzymes (*Chen et al., 2024*).

SHV-1 is a very well-characterized enzyme with a wealth of information on mutations and their corresponding effects on protein function. In contrast, the information available on PDC-3 remains limited. Essential catalytic residues in SHV-1 are $S_{70}$, $K_{73}$, $S_{130}$, $E_{166}$, $N_{170}$, $K_{234}$, $G_{236}$, and $A_{237}$ (*Ambler et al., 1991*), and conserved catalytic residues in PDC-3 include $S_{64}$, $K_{67}$, $Y_{150}$, $N_{152}$, $K_{315}$, $T_{316}$, and $G_{317}$. Highly conserved stretches of 3–9 hydrophobic residues, annotated as hydrophobic nodes, exist in class A β-lactamases and have been proven to be essential for protein stability (*Galdadas et al., 2018*). Residues defined as belonging to hydrophobic nodes within SHV-1 are listed in *Supplementary file 1a*.

In SHV-1, the predominant extended spectrum β-lactamase (ESBL) substitutions occur at $L_{35}$, $G_{238}$, and $E_{240}$, while $R_{43}$, $E_{64}$, $D_{104}$, $A_{146}$, $G_{156}$, $D_{179}$, $R_{202}$, and $R_{205}$ appear in ESBLs with lower frequency (*Liakopoulos et al., 2016*). Mutations at $M_{69}$, $S_{130}$, $A_{187}$, $T_{235}$, and $R_{244}$ are known to induce inhibitor resistance in the enzyme (*Pagan-Rodriguez et al., 2004*). In PDC-3, substitutions primarily occur on the Ω-loop, enhancing its flexibility to accommodate the bulky side chains of antibiotics, while deletions are more common in the R2-loop (*Jacoby, 2009*). The predominant Ω-loop mutations isolated from clinics are found at positions $V_{211}$, $G_{214}$, $E_{219}$, and $Y_{221}$ (*Barnes et al., 2018*).

## Emergence of highly conserved dynamic couplings

DyNoPy builds a pairwise model of conserved dynamic couplings detected by combining coevolution scores and information on functional motions into a score $J_{ij}$ (see 'Methods' and *Figure 1*). To this end, a dynamic descriptor should be selected. When the descriptor is associated with functional conformational changes, it is expected that functionally relevant couplings will report higher scores. Dynamic descriptors can be selected from commonly used geometrical collective variables (CVs) for the analysis of MD trajectories (see 'Methods'). As expected, the average $J$ matrix score varies across the different CVs, with some of them showing no signal of dynamic coupling (*Figure 3A*).

SHV-1 and PDC-3 exhibit distinct dynamics, requiring a different choice of the CV that best captures the functional dynamics. For SHV-1, the global first principal component (PC1) proved to be the most effective feature, identifying 571 residue pairs with a $J_{ij}$ value greater than 0. Conversely, PDC-3 requires selection of more localized features that can extract the Ω-loop dynamics from the overall protein motion. Among the dynamic descriptors, the partial first time-lagged component (TC1_partial) performed best for PDC-3, detecting 216 residue pairs with a $J_{ij}$ value greater than 0. Consequently, PC1 and TC1_partial were selected to build the $J$ matrix for SHV-1 and PDC-3, respectively. The performance of all 12 CVs for each protein was assessed and listed in *Supplementary file 1b*.

The importance of dynamical information is evident when coevolution couplings ($\gamma_{ij}$) and conserved dynamic couplings ($J_{ij}$) are compared: the number of non-zero couplings decreases from 40% to <2% of total residue pairs in the protein (*Figure 3B*) when information from the dynamics descriptor is added. Thus, the inclusion of protein dynamics in coevolution studies acts as an effective filter that rules out residue pairs that do not have significant correlations with functional motions. Moreover, when relying only on $\gamma_{ij}$, all the residues in SHV-1 and PDC-3 are included within four identified communities (*Supplementary file 1c*), suggesting that coevolution scores ($\gamma_{ij}$) alone do not effectively discriminate residues relevant for protein functions. Furthermore, it would be hard to distinguish critical core residues for each community using only $\gamma_{ij}$ as the eigenvector centrality (EVC) values for the residues do not show remarkable differences (*Figure 3—figure supplement 1A and B*). This means that detailed dynamic investigation of the top residues is needed to determine which pairs should be picked up and further analysed. On the other hand, it is much easier to identify essential residues based on $J$ scores calculated as clear outliers with significantly higher EVC values could be seen for

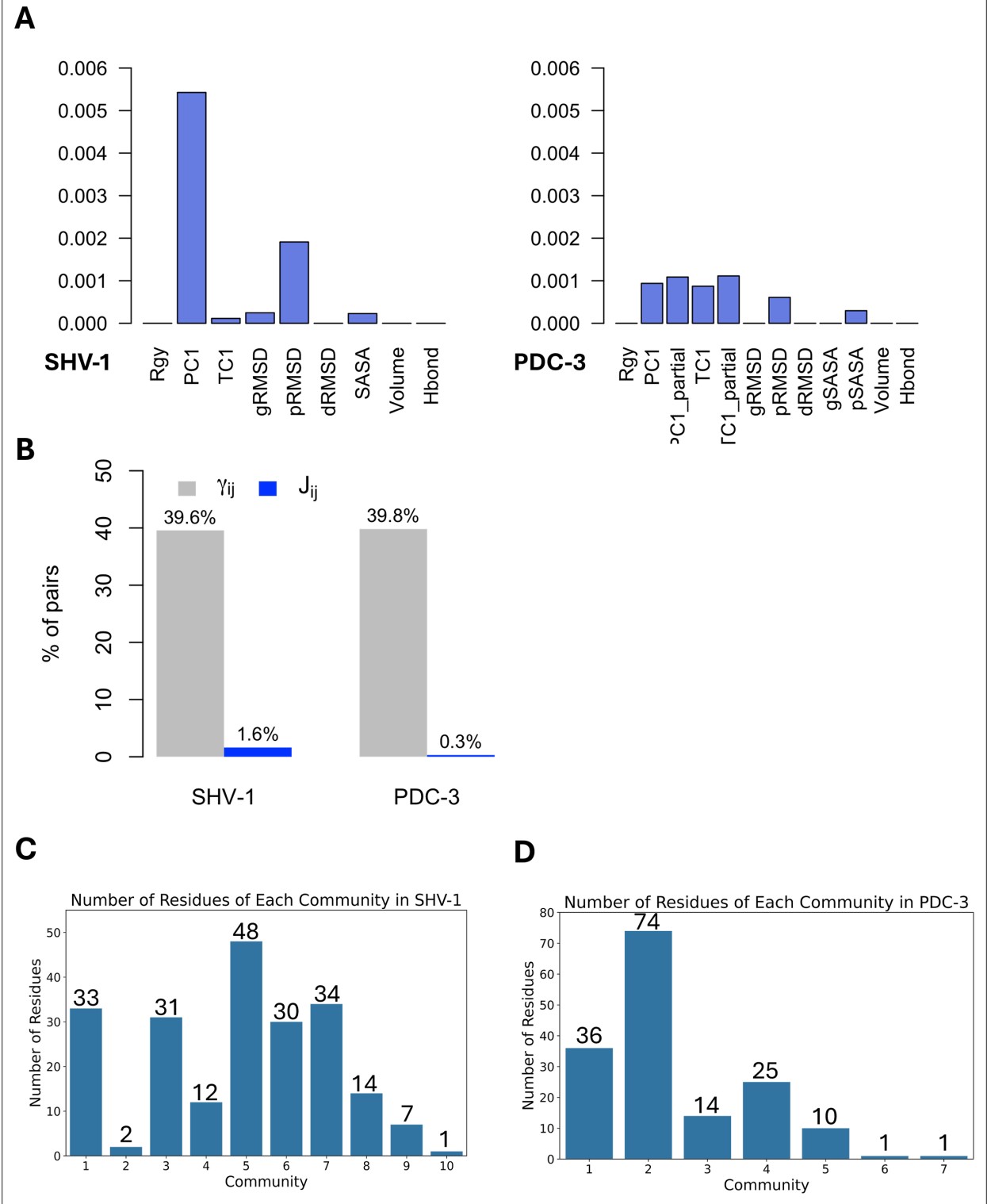

**Figure 3.** Features and residues in each community. (**A**) Average $J$ matrix score varies across different collective variables (CVs). Left: SHV-1; right: PDC-3. (**B**) Number of non-zero couplings detected by scaled coevolution scores ($\gamma_{ij}$) and $J$ values calculated by DyNoPy ($J_{ij}$). (**C**) Number of residues of each community in SHV-1. (**D**) Number of residues of each community in PDC-3. A reasonable residue community should contain at least three residues.

The online version of this article includes the following figure supplement(s) for figure 3:

**Figure supplement 1.** Eigenvector centrality (EVC) distribution of all the residues for each community.

almost all communities (*Figure 3—figure supplement 1C and D*; *Parente et al., 2015*; *Negre et al., 2018*). In conclusion, the lack of specificity in the statistically based coevolution analysis supports the choice of incorporating a score for the correlation between residue interactions and dynamic behaviours that enables deconvolution of community information.

## DyNoPy reveals critical residues and predicts evolutionary pathways in SHV-1

DyNoPy identified eight meaningful communities, each consisting of at least three strongly coupled residues within SHV-1 (*Figure 3C*). All crucial catalytic residues and critical substitution sites previously mentioned participating in one of these communities with the exceptions of $R_{43}$, $R_{202}$, and $S_{130}$. Residues previously known to have critical role in function or conferring ESBLs/IRBLs phenotype are either directly coupled to protein dynamics or act as a central hub. The hubs interact with residues with either a role in catalysis or structural stability through their membership of hydrophobic nodes (*Olehnovics et al., 2021*). Furthermore, DyNoPy identified key positions ($L_{162}$ and $N_{136}$) within some communities that are known to undergo substitutions, conferring an ESBL phenotype in other class A β-lactamases. These substitutions have not yet emerged in the SHV family, providing insightful predictions about the potential future evolution of the enzyme. Detailed discussions of communities with secondary importance for protein function (communities 3, 8, and 9) is provided in the Appendix (*Appendix 2—figure 1*).

## DyNoPy predicts mutation hotspots in SHV-1

DyNoPy detects critical mutation sites ($L_{162}$ and $N_{136}$) that are known to extend the range of substrates in other class A β-lactamases but have not yet emerged as variants in the SHV family. These sites have not been modified in the SHV family because of their plausible central role within the communities as they are mediating couplings with key functional residues essential for catalytic activity and structural stability, indicating their critical role in protein function and the potential lower mutation rate. These findings provide insightful predictions about the potential future evolution of the enzyme, as well as plausible explanations for why these mutations have not yet appeared.

$L_{162}$, positioned at the start of the Ω-loop and adjacent to the crucial catalytic residue $E_{166}$, is assigned as the core residue for community 1 (*Figure 4A*). While it remains conserved in the SHV family, variants of $L_{162}$ have been isolated in other class A β-lactamase and are known to expand the enzyme catalytic spectrum. Single amino acid substitution at $L_{162}$ can intensify antibiotic resistance in BEL-1 (*Pozzi et al., 2016*), a class A ESBL clinical variant, exhibiting robust resistance to ticarcillin and ceftazidime (*Bogaerts et al., 2007*). BEL-2 diverges from BEL-1 by single amino acid substitution ($L_{162}F$), which alters the kinetic properties of the enzyme significantly and increases its affinity towards expanded-spectrum cephalosporins (*Poirel et al., 2010*). The relationship between $L_{162}$ and protein catalytic functions can be explained using DyNoPy model as there are couplings with catalytic important residues $M_{69}$, $K_{73}$, $E_{166}$, and $K_{234}$. Moreover, the BEL case has confirmed that $L_{162}F$ mutation significantly destabilizes the overall protein structure, highlighting the crucial role of $L_{162}$ in maintaining protein stability (*Pozzi et al., 2016*). DyNoPy accurately identifies the centrality of $L_{162}$ by reporting its connections with 28 backbone residues, including 9 hydrophobic node residues critical for protein stability. Among these, five hydrophobic residues are part of the α2 node: $V_{75}$, $L_{76}$, $G_{78}$, $V_{80}$, and $L_{81}$, highlighting the contribution of $L_{162}$ to the stability of the α2 helix (*Olehnovics et al., 2021*).

Just like $L_{162}$, $N_{136}$ undergoes advantageous mutations in other class A β-lactamases while remains highly conserved within the SHV family. It is the core residue for community 7 (*Figure 5B*). This residue forms a hydrogen bond with $E_{166}$, stabilizing the Ω-loop (*Bös and Pleiss, 2008*). Although DyNoPy did not detect this direct interaction between $N_{136}$ and $E_{166}$, the established relationship between $N_{136}$ and $N_{170}$ highlights the role of $N_{136}$ in influencing $E_{166}$. $N_{170}$, an essential catalytic residue located on the Ω-loop, contributes to priming the water molecule for the deacylation step with $E_{166}$ (*Agarwal et al., 2023*), and is directly coupled with $N_{136}$. Due to the essential contribution of $N_{136}$ in facilitating $E_{166}$ to maintain its proper orientation, it was previously thought to be intolerant to mutations as substitution of asparagine to alanine at this position would make the enzyme lose its function completely (*Cao et al., 2020*). However, $N_{136}D$ substitution has emerged as a new clinical variant very recently in PenL, a class A β-lactamase, by increasing its ability in hydrolysing ceftazidime (*Cao et al., 2020*), suggesting that this site has potential to mutate. This gain of function is mainly triggered by the increased flexibility of the Ω-loop (*Cao et al., 2020*). DyNoPy correctly detects a dynamical relationship between

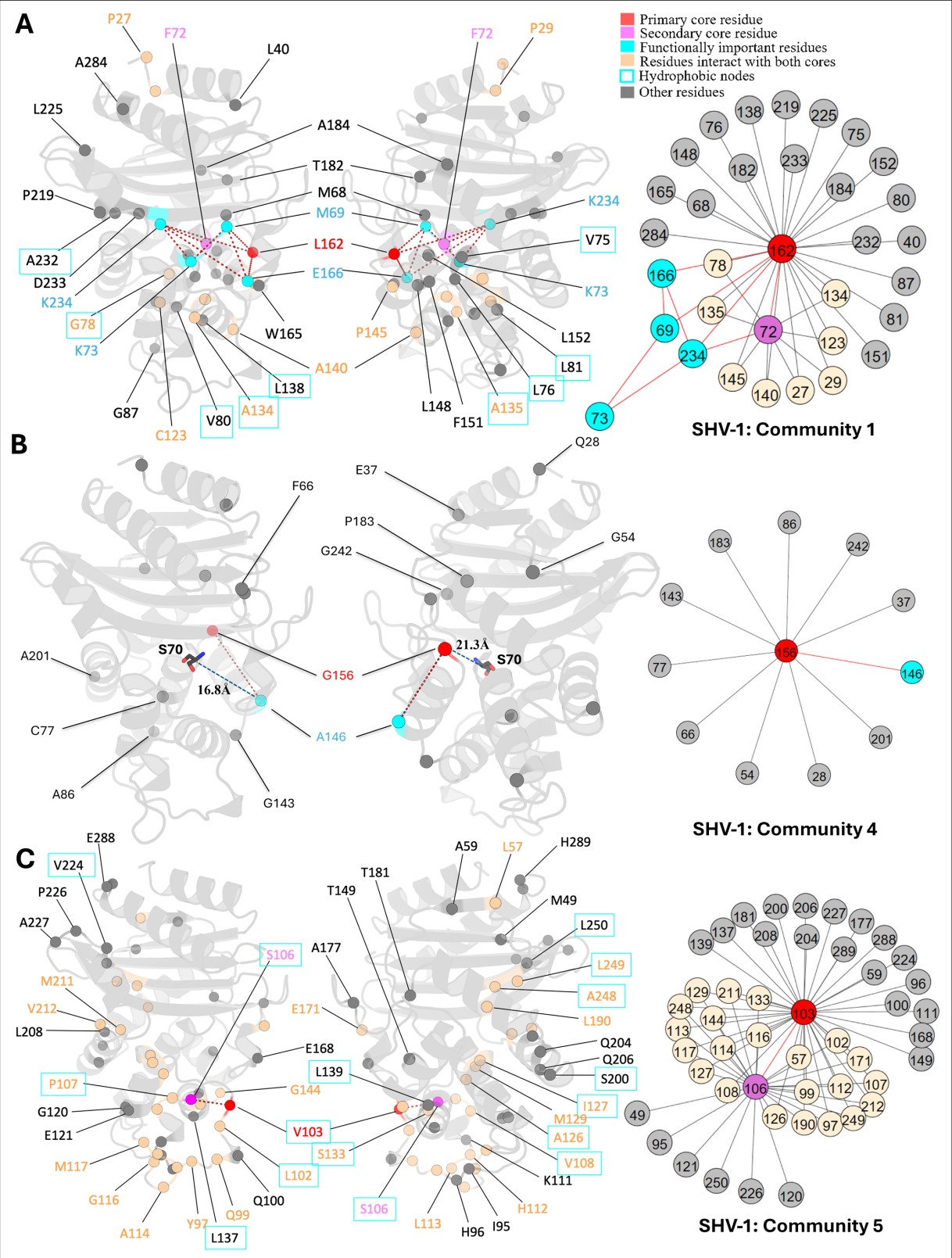

**Figure 4.** Communities 1, 4, and 5 of SHV-1 β-lactamase. All the residues are depicted as spheres on the protein structure. The core residue for each community is highlighted in red, while purple is used to emphasize the secondary core residue. Residues that interact with both cores are coloured in light yellow. Functional important residues are marked in cyan. Hydrophobic nodes are enclosed with cyan boxes. (**A**) Community 1 of SHV-1, comprising 33 residues with $L_{162}$ being the primary core residue. (**B**) Community 4 of SHV-1, containing 12 residues and is centred by $G_{156}$. $G_{156}$ and $A_{146}$

*Figure 4 continued on next page*

*Figure 4 continued*

are two functional important residues distant from the active site. $G_{156}$ is 21.3 Å away from the catalytic $S_{70}$. $A_{146}$ is 16.8 Å away from $S_{70}$. (**C**) Community 5 of SHV-1, embracing 48 residues and showing a strong correlation between $V_{103}$ and $S_{106}$.

$N_{136}$ and the Ω-loop (residues 164–179). Six residues present in the Ω-loop participate within this community, including $R_{164}$ and $D_{179}$. These two residues are critical as they are forming the 'bottleneck' of the Ω-loop which is essential for the correct position of $E_{166}$ (*Parwana et al., 2024*). $D_{179}$ is also a critical mutation site for SHV-1. Single amino acid substitutions like $D_{179}A$, $D_{179}N$, and $D_{179}G$ are enough for the extended spectrum phenotype (*Liakopoulos et al., 2016*).

## DyNoPy detects residue couplings essential for protein stability

DyNoPy identifies residue couplings critical for protein functional motions, particularly associated with protein stability. These residue pairs exhibit strong relationships as they are not only directly coupled with each other but also forms various indirect couplings via other residues. As a result, both

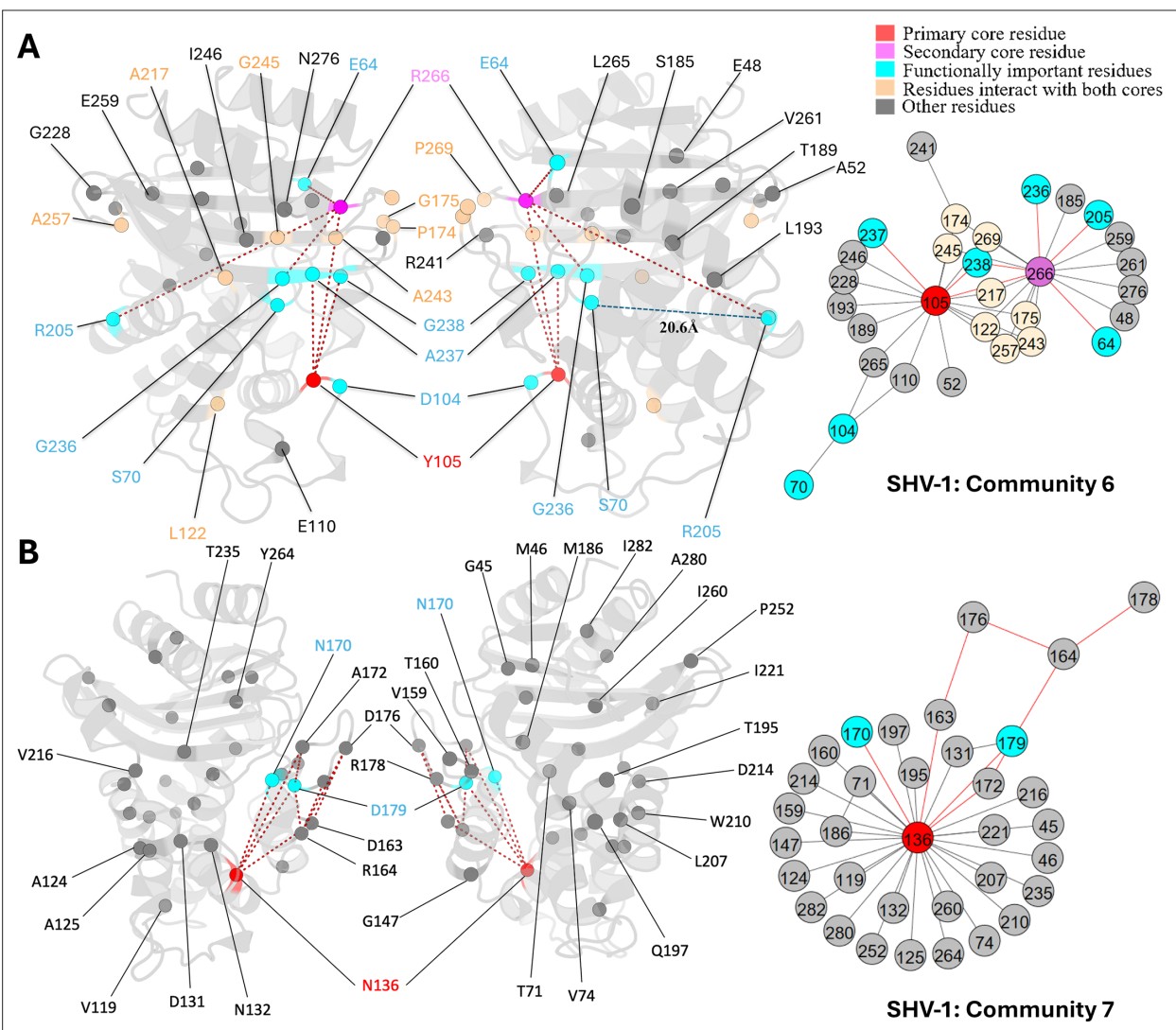

**Figure 5.** Communities 6 and 7 of SHV-1 β-lactamase. All the residues are depicted as spheres on the protein structure. The core residue for each community is highlighted in red, while purple is used to emphasize the secondary core residue. Residues that interact with both cores are coloured in light yellow. Functional important residues are marked in cyan. (**A**) Community 6 of SHV-1, comprising 30 residues with $Y_{105}$ being the primary core residue. $R_{205}$ is a functional important residue that is 20.6 Å away from the active site $S_{70}$. (**B**) Community 7 of SHV-1, containing 34 residues and is centred by $N_{136}$.

residues are considered as core residues inside these communities. It is expected that disruption of these couplings through mutation could compromise collective motions essential for enzyme activity.

As the secondary core residues in community 1 (*Figure 4A*), $F_{72}$ is showing a strong coupling with the primary core residue $L_{162}$ and also forms nine indirect couplings with $L_{162}$, including via the catalytic $K_{234}$. This network of direct and indirect relationships reveals the importance of $F_{72}$ and $L_{162}$ coupling in maintaining protein functional motions. Interestingly, previous studies identified a small hydrophobic cavity formed by $L_{162}$ and $F_{72}$, together with $L_{139}$, and $L_{148}$, which is essential for the stability of the active site (*Pozzi et al., 2016*). Notably, DyNoPy successfully recovers the key residues of this local hydrophobic cavity ($L_{162}$, $F_{72}$, and $L_{148}$).

The strong interplay between $V_{103}$ and $S_{106}$, which are both residues on the α3-α4 loop, is seen in community 5 (*Figure 4C*). These residues not only interact with each other directly but are also indirectly coupled via 22 other residues. This community emphasizes the significance of hydrophobic nodes in SHV stability and dynamics. Within the analysed 48 residues, 27 are hydrophobic, out of which 15 residues act as nodes critical for enzyme stabilization. Hydrophobic nodes stabilize their own secondary structures and interconnect to stabilize the overall protein (*Galdadas et al., 2021*). $V_{103}$ and $S_{106}$ themselves are hydrophobic nodes, stabilizing α3 helix and α4 helix, respectively, and are strongly coupled with each other. In CTX-M, another class A enzyme, $N_{106}S$ is a common substitution that results in improved thermodynamic stability and compensate for the loss in stability of the variants (*Lu et al., 2022*). Interestingly, this residue is already a serine in SHV but still implies its pivotal role in protein stability.

## DyNoPy provides valid explanations for mutation sites

During the evolution of β-lactamases, single mutations on specific sites that are distant from the functional sites have been observed to significantly alter protein catalytic functions. Additionally, single mutations on some surface exposed residues can dramatically increase protein stability. Understanding how these distant mutations impact function and stability becomes a major challenge in understanding protein evolutionary pathways. Communities extracted by DyNoPy show these residues linked with functional important residues, providing a rational for these mutation sites with unknown functions.

Mutations of $G_{156}$ are limited but they lead to ESBL phenotype in the SHV family (*Liakopoulos et al., 2016*). $G_{156}$ is the central residue for community 4 (*Figure 4B*), but it is distant from the active site, over 20 Å away from the catalytic serine $S_{70}$. Clinical variant SHV-27 has extended resistance ability towards cefotaxime, ceftazidime, and aztreonam (*Corkill et al., 2001*). It differs from SHV-1 by single amino acid substitution $G_{156}D$, suggesting that it has directly evolved from SHV-1 (*Corkill et al., 2001*). Limited research has been done on position $G_{156}$, and the understanding of how it affects the enzyme catalytic properties given that it is far away from the active site is still unclear. Based on our results, we suggest that this residue is essential for the overall protein function because of its 11 coevolved dynamic couplings with protein dynamics, including $A_{146}$, another ESBL substitution site.

SHV-38, another ESBL that is capable of hydrolysing carbapenems, harbours a single $A_{146}V$ substitution compared to SHV-1 (*Poirel et al., 2003*). Like $G_{156}$, $A_{146}$ is 16.8 Å away from $S_{70}$ but shows an ability in altering protein catalytic function. The $A_{146}$-$G_{156}$ residue pair shows a strong coevolutionary signal and strong correlation with protein overall dynamics, implying that there may compensatory mutations at these sites with potential to emerge in the SHV family in the future. These two residues are not connected to any catalytic residues but their coupling to functional dynamics can offer plausible explanation to ESBL activity of these two mutations.

Unlike other substitution sites that are adjacent to the active site, $R_{205}$ is situated more than 20 Å away from catalytic serine $S_{70}$. Its side chain points outwards from the protein, exposing to the solvent. The $R_{205}L$ substitution often co-occurs with other ESBL mutations and is thought to indirectly contribute to the ESBL phenotype by compensating for stability loss induced by other mutations (*Ben Achour et al., 2009*). SHV-3 is an ESBL that exhibits significant resistance to cefotaxime and ceftriaxone (*Nicolas et al., 1989*). Two substitutions in this enzyme, $R_{205}L$ and $G_{238}S$, extend its resistance profile (*Nicolas et al., 1989*). Thus, it is promising to see that DyNoPy detected these two mutation sites together within community 6 (*Figure 5A*).

$Y_{105}$ and $R_{266}$ are the core residues for community 6 (*Figure 5A*). $Y_{105}$ is situated on the α3-α4 loop positioned at the left side of the binding pocket. It is an important catalytic residue that recognizes and binds to the thiazolidine ring of penicillins or β-lactamase inhibitors (*Bethel et al., 2006*). There

is very limited information on the role of $R_{266}$, except that it may stabilize the Ω-loop in the SHV family similar to the analogous $T_{266}$ in TEM (*Kuzin et al., 1999*). $G_{238}$ is coupled with an essential catalytic residue $Y_{105}$, which further links with other catalytic functional residues: $S_{70}$ and $A_{237}$, and $R_{266}$, a residue that is known to stabilize the Ω-loop. This indicates that mutations on $G_{238}$ would result in an alteration on protein catalytic function, as well as an increased flexibility of the protein, which strongly aligns with previous finding (*Nicolas et al., 1989*). Its linked mutation site $R_{205}$ does not show direct coupling with any catalytic residues. Instead, it is directly coupled with $R_{266}$, which we mentioned as an Ω-loop stabilizer. Thus, it is not surprising that $R_{205}$ substitution alone is never observed in nature (*Neubauer et al., 2020*) as it would not give significant evolutionary advantage to the protein.

## Insights into the unexplained functional sites of PDC-3

Unlike the extensively studied SHV-1, the functional roles of individual amino acids in PDC-3 remain largely unexplored. This gap in understanding serves as welcome challenge for interpreting the effects of mutations and the dynamic behaviour of PDC-3 from our results. Although several mutation hotspots, such as those on the Ω-loop (*Barnes et al., 2018*), have been identified, very little is known about the specific contributions of individual amino acids on the functionality of PDC-3.

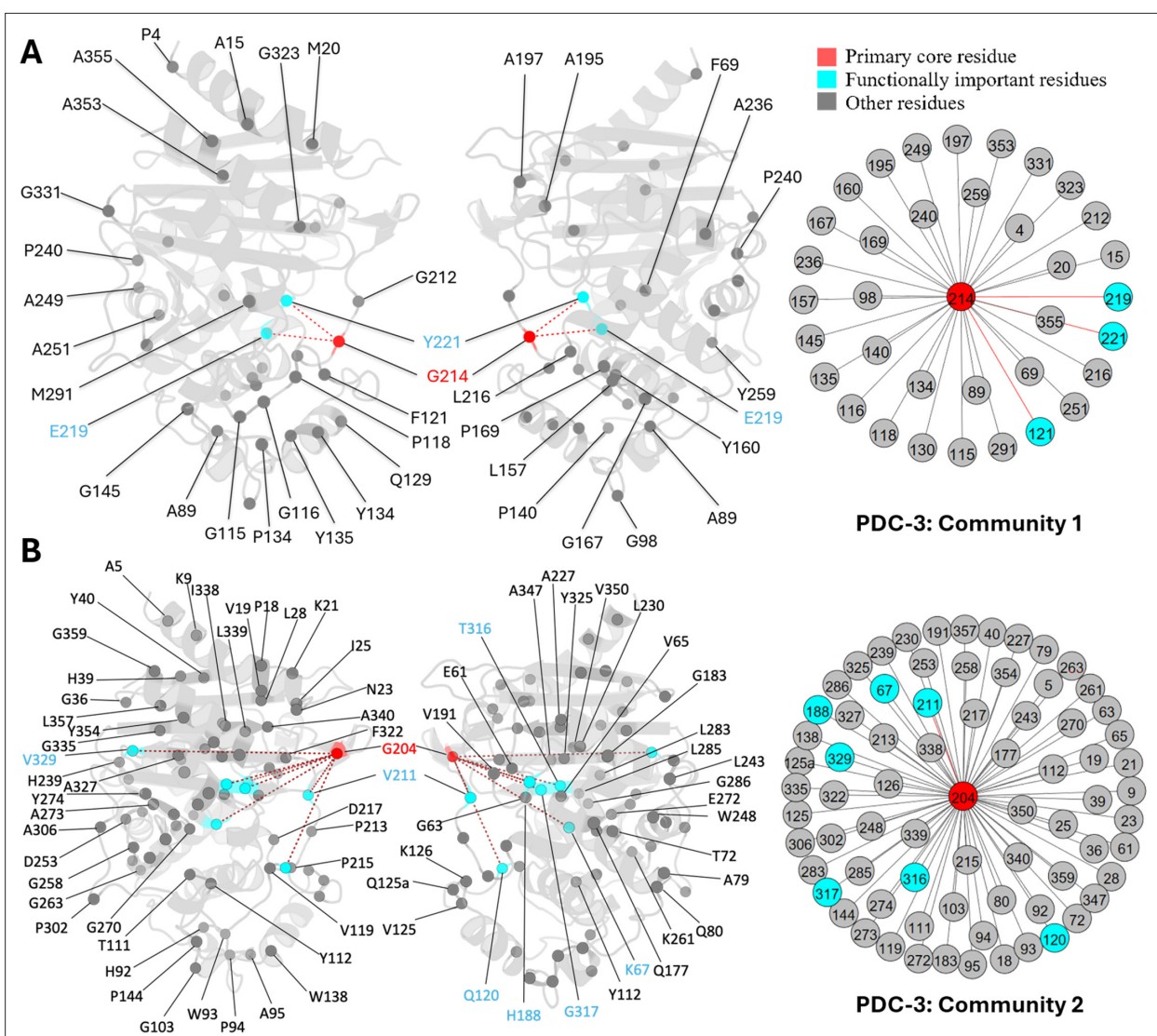

**Figure 6.** Communities 1 and 2 of PDC-3 β-lactamase. All the residues are depicted as spheres on the protein structure. The core residue for each community is highlighted in red. Functional important residues are marked in cyan. (**A**) Community 1 of PDC-3, comprising 36 residues with $G_{214}$ being the primary core residue. (**B**) Community 2 of PDC-3, containing 74 residues and is centred by $G_{204}$.

In PDC-3, mutations have primarily been reported in the Ω-loop. They enhance its flexibility to accommodate the bulky side chains of antibiotics, while deletions are more common in the R2-loop (*Jacoby, 2009*). DyNoPy detected five communities in total (*Figure 3D*), with all the four predominant Ω-loop mutations appearing in these communities. Communities 3, 4 and 5 are discussed in the Appendix (*Appendix 2—figure 2*). Furthermore, DyNoPy also detected several previously unexplored Ω-loop residues.

$G_{214}$, a known mutation site in PDC-3, is the core residue in community 1. Another two essential mutation sites, $E_{219}$ and $Y_{221}$, also participate in this community, directly coupled with $G_{214}$ (*Figure 6A*). $G_{214}$ also has direct couplings with four other Ω-loop residues: $A_{195}$, $A_{197}$, $G_{212}$, and $L_{216}$. Previous results have demonstrated that substitutions of glycine to alanine or arginine at 214 significantly destabilize the Ω-loop (*Chen et al., 2024*). The strong correlation between $G_{214}$ and these Ω-loop residues emphasizes the significant contribution of $G_{214}$ towards the stability of the Ω-loop, which corroborates with previous results (*Chen et al., 2024*). Moreover, substitutions such as $G_{214}A$ and $G_{214}R$ and mutations on $E_{219}$ and $Y_{221}$ do not affect R2 loop flexibility, resulting in the smaller active site volume among variants (*Chen et al., 2024*) because none of the residues from the R2 loop are detected in this community, offering plausible explanation to the previously unexplained phenomenon.

$G_{204}$ is the core residue of community 2, coupled with 73 other residues, most of which are distant from the catalytic site, suggesting plausible crucial role in the overall protein stability like $L_{162}$ in SHV-1 (*Figure 6B*). $G_{204}$, a newly emerged mutation site in the PDC family (67), is located on the short β-sheet β5a within the Ω-loop, near the hinge region between β8 and β9 just above the active site. The only known variant of $G_{204}$ is PDC-466, which was derived from PDC-462 ($A_{89}V$, $Q_{120}K$, $V_{211}A$, $N_{320}S$), with an addition of $G_{204}D$ (*Colque et al., 2021*). Coupling of $G_{204}$ to several catalytically important residues, including $K_{67}$, $K_{315}$, and $T_{316}$ can suggest that mutations at this site can negatively impact catalytic power. This offers a plausible explanation of seeing fewer variants at this site, and mutations at this site could have an impact on the hydrolysing capabilities of PDC variants. This should be confirmed by further experimental studies of variants of $G_{204}$. Unlike $G_{214}$, $E_{219}$, and $Y_{221}$ mutations which do not influence the dynamics of the R2 loop, substitutions on $V_{211}$, a member of Ω-loop, have an impact on the dynamics of the R2 loop because of its indirect couplings, through $G_{204}$ to R2-loop residues (*Chen et al., 2024*). Two less critical substitution sites, $H_{188}$ and $V_{329}$, were also observed in community 2.

## Conclusions

DyNoPy offers two distinct advantages over existing computational tools (*Yehorova et al., 2024*; *Osuna, 2021*): (a) information on residue–residue coevolution can be directly used to detect the components of protein dynamics that have been preserved during evolution and (b) dynamic descriptors extracted from the MD ensembles can be used to identify the function-specific conserved dynamic couplings. These couplings are then easily modelled as a graph, and network analysis is used to extract epistatic communities and assign roles to residues based on their importance in the graph model. The choice of a relevant descriptor of functional dynamics has an impact on the ability to detect couplings that are involved in functional dynamics.

Here we demonstrated how the choice of relevant global and local descriptors returns a higher number of effective couplings (greater than 0), and in turn leads to interpretable graph models and communities. In other systems, when multiple descriptors can be used to quantify functional conformational change, it is expected that they will differently modulate the effect of coevolution coupling, which will be reflected in a different structure of the associated graph models. This suggests the use of DyNoPy to generate comparative models in proteins with multiple functions associated to distinct dynamical changes.

Mutations of $L_{162}$ and $N_{136}$ have not yet emerged in SHV-1, but they are detected by DyNoPy as core residues for communities. These residues are strongly coupled with other functional important residues, which play critical roles in protein stability and catalytic activity. The identification of these couplings shows high consistency with previous studies and highlights the importance of $L_{162}$ and $N_{136}$ in SHV-1 functional dynamics. Given their central role in these communities, mutations in $L_{162}$ and $N_{136}$ can significantly alter protein function, suggesting their potential for future evolutionary changes. However, their strong relationships with these critical functional residues also suggest that mutation at these sites would need to be balanced to maintain protein function, providing an explanation for why such mutations have not yet emerged in SHV-1 (*Soskine and Tawfik, 2010*). The ability of DyNoPy

in detecting functionally important mutation sites was demonstrated via well-characterized mutation sites including $R_{205}$ and $G_{238}$ from SHV-1. Moreover, DyNoPy shows predictive ability on less-studied mutation sites such as $G_{156}$ and $A_{146}$, by detecting critical residue couplings that coevolved with functional motions.

Based on the knowledge we have gained from the analysis of SHV-1 functional protein dynamics, we suggest that in PDC-3, mutations at $G_{204}$ because of its significantly conserved dynamic couplings can lead to new ESBL/IRBL clinical variants. We suggest that DyNoPy can be used as a predictive tool to identify potential functional residues within this enzyme and guide future mutagenesis studies.

In summary, by integrating hidden evolutionary information with direct dynamic interactions, DyNoPy provides a powerful framework for identifying and analysing functional sites in proteins. The tool not only identifies key residues involved in local and global interactions but also improves our ability to predict silent residues with previously unknown roles for future experimental testing. Our application of DyNoPy to broad-spectrum β-lactamases ESBLs and IRBLs demonstrates its potential to address key medical challenges such as antibiotic resistance by providing valid predictions on protein evolution.

## Methods

DyNoPy generates a graph representation of the protein structure that captures the couplings between amino acid residues contributing to the functional dynamics of the protein. Residues are represented as graph nodes, and conserved dynamic couplings are recorded as edges. Edge weights quantify the strength of these couplings. The model is built on two assumptions: residue pairs should have (a) coevolved and their (b) time-dependent interactions correlate with a functional conformational change.

Therefore, edge weights ($J_{ij}$) for residue $i$ and $j$ are calculated as

$$J_{ij} = \alpha\gamma_{ij} + \beta\rho_{ij} \ where \alpha + \beta = 1 \tag{1}$$

where $\gamma_{ij}$ is the scaled coevolution score and $\rho_{ij}$ is the degree of correlation with the selected functional conformational change. α and β are weights assigned to $\gamma_{ij}$ and $\rho_{ij}$ that have a sum of 1. The relative weight of the scaled coevolution score (α) is set to 0.5 in this study. When either of the assumptions listed above is not met, $J_{ij}$ is set to zero.

### Scaled coevolution scores

The occurrence of residue–residue coevolution can be estimated and quantified using probabilistic models of correlated mutations from deep MSA. DyNoPy supports generation of the MSA using the HH-Suite package (*Remmert et al., 2011*) and calculation of scaled coevolution score ($\gamma_{ij}$) using CCMpred (*Seemayer et al., 2014*) as per the protocol described in *Bibik et al., 2024*. For SHV-1 and PDC-3, hhblits returned 18,174 sequences ($N_{eff}$: 11.082) and 27,892 sequences ($N_{eff}$: 9.951). Sequences were detected from the UniRef30 (v2022_02) database (*Mirdita et al., 2017*). First, a pairwise residue coevolution matrix (**C**) is calculated, then these raw scores ($c_{ij}$) are divided by the matrix mean (*Equation 2*). All scores ($s_{ij}$) smaller than 1 are set to zero, and the remaining values are normalized by the maximum value (*Equation 3*):

$$s_{ij} = \frac{c_{ij}}{\langle C \rangle} \tag{2}$$

$$\gamma_{ij} = \begin{cases} 0, & s_{ij} < 1 \\ \frac{s_{ij}}{s_{max}}, & s_{ij} \geq 1 \end{cases} \tag{3}$$

### Correlation with functional motions

The contribution of a residue pair to a selected functional motion is estimated by how much the change in interaction energy between the two residues over time is correlated with a CV describing the functional motion:

$$r_{ij} = cor\left(\varepsilon_{ij}\left(t\right)_d\left(t\right)\right) \tag{4}$$

$$\rho_{ij} = \begin{cases} 0, & r_{ij} \leq 0.5 \\ r_{ij}, & r_{ij} > 0.5 \end{cases} \tag{5}$$

where $\varepsilon_{ij}(t)$ is the pairwise non-bonded interaction energy (see details in Appendix 1) and $d(t)$ is the time-dependent value of the CV. Examples of CV and a discussion on the choice of the most relevant CV are presented in the 'Results' section. Correlation values smaller than 0.5 are set to 0. In the absence of detectable contributions to the functional dynamics of the system, the couplings extracted by DyNoPy will describe a pure evolutionary model, and the community detection method presented below will be equivalent to a direct decomposition of the residue coevolution network into units.

## Graph representation and analysis of conserved dynamic couplings

All pairwise conserved dynamic couplings (*Equation 1*) are collected into a square matrix *J*. A graph is built from *J* using python-igraph v0.11 library (*Csárdi and Nepusz, 2006*). Nodes represent residues, and edges are drawn between nodes with positive $J_{ij}$. Edge weights are set to $J_{ij}$. The relative importance of the residues in this model of protein dynamics is calculated as EVC of the nodes (*Newman, 2004*). The residues involved in extensive correlated dynamics with other highly connected residues have higher EVC scores. Groups of residues contributing to important collective motions are detected by community analysis of the graph structure. The Girvan–Newman algorithm is used to extract the community structure (*Newman, 2006*). A meaningful community should contain at least three residues. Applying network analysis on the combined dynamics-coevolution matrix helps us extract higher-order interactions beyond pairwise coupling and detecting critical residues, which show multiple interactions with each other. Moreover, indirect long-range relationships, which would be hard to identify from numerical data, could be detected through community clustering. Community-based analysis offers a more comprehensive understanding of residue relationships and enables the visualization of residue couplings on the protein structure.

## Adaptive sampling molecular dynamics simulations

MD simulation data was sourced from our previous studies (*Olehnovics et al., 2021*; *Chen et al., 2024*). To summarize, SHV-1 structural coordinates (PDB ID: 3N4I) were obtained from the Protein Data Bank and modified to the wild type by introducing the E104D mutation. Similarly, the PDC-3 structure was derived from PDC-1 (PDB ID: 4HEF) by a T105A substitution. Both enzymes were protonated at pH 7.0 using PropKa from the PlayMolecule platform (*Martínez-Rosell et al., 2017*). One disulfide bond between $C_{77}$ and $C_{123}$ was specified in SHV-1. Both structures were solvated with TIP3P water molecules in a periodic box with a box size of 10 Å. Ions were added to neutralize the overall charge of each system at 150 mM KCl. Amber force field ff14SB was used for all MD simulations (*Maier et al., 2015*). After an initial minimization of 1000 steps, both the enzymes were equilibrated for 5 ns in the NPT ensemble at 1 atmospheric pressure using the Berendsen barostat (*Berendsen et al., 1984*). The initial velocities for each simulation were sampled from the Boltzmann distribution at 300 K. Multiple Markov state model (MSM)-based adaptively sampled simulations were performed for both proteins based on the ACEMD engine (*Doerr et al., 2016*; *Harvey et al., 2009*). A canonical (NVT) ensemble with a Langevin thermostat (*Davidchack et al., 2009*) (damping coefficient of 0.1 ps−1) and a hydrogen mass repartitioning scheme were employed to achieve time steps of 4 fs. For SHV-1, each trajectory spanned 60 ns with a time step of 0.1 ns, with a total of 593 trajectories. In the case of PDC-3, 100 trajectories were collected, each containing 3000 frames, lasting 300 ns. To manage the extensive datasets efficiently, trajectories were strategically stridden to ensure that a minimum of 30,000 frames were preserved for each system. The resulting trajectories are summarized in *Supplementary file 1d*.

## Calculation and selection of collective variables

DyNoPy works on the assumption that time-dependent interactions between critical residues, either having significant structural change or not, will correlate with functional conformational motions. Since MD simulation data is high-dimensional, a time-dependent CV is required to extract the most relevant information for the process under study. The usefulness of DyNoPy is dependent on the choice of the CVs. To guide the selection of CVs, we selected 12 distinct features: radius of gyration ($R_g$), the first

principal component (PC1), partial PC1 (PC1_partial), the first time-lagged independent component (TC1), partial TC1 (TC1_partial), global root mean square deviation (gRMSD), partial RMSD (pRMSD), dynamical RMSD (dRMSD), global solvent-accessible surface area (gSASA), partial SASA (pSASA), active site pocket volume, and the number of hydrogen bonds (hbond). A description of the CVs, including the calculation methods and the residues used to calculate the partial variables, is detailed in Appendix 1. CVs were subsequently used as input features for DyNoPy. A good CV should appropriately describe protein functional motions. Thus, a CV that detects the highest number of residue couplings is expected to be the most suitable descriptor. The length of the MD simulations should be appropriate to effectively sample the desired functional process as described by the selected CV.

## Acknowledgements

SCD was supported by Leverhulme Trust grant RPG-2017-222 awarded to AP and JAG. The authors would like to thank Arianna Fornili for insightful suggestions on the design of DyNoPy methodology.

## Additional information

### Competing interests

Shozeb Haider: Reviewing editor, eLife. The other authors declare that no competing interests exist.

### Funding

| Funder | Grant reference number | Author |
| --- | --- | --- |
| Leverhulme Trust | RPG-2017-222 | James A Garnett<br>Alessandro Pandini |

The funders had no role in study design, data collection and interpretation, or the decision to submit the work for publication.

### Author contributions

Manming Xu, Formal analysis, Validation, Investigation, Visualization, Writing - original draft; Sarath Chandra Dantu, Conceptualization, Software, Formal analysis, Validation, Investigation, Visualization, Methodology, Writing - original draft, Writing – review and editing; James A Garnett, Conceptualization, Investigation, Writing – review and editing; Robert A Bonomo, Validation, Project administration, Writing – review and editing; Alessandro Pandini, Conceptualization, Software, Formal analysis, Supervision, Validation, Investigation, Visualization, Methodology, Project administration, Writing – review and editing; Shozeb Haider, Conceptualization, Resources, Formal analysis, Supervision, Validation, Investigation, Visualization, Methodology, Project administration, Writing – review and editing

### Author ORCIDs

Sarath Chandra Dantu (iD) https://orcid.org/0000-0003-2019-5311
Shozeb Haider (iD) https://orcid.org/0000-0003-2650-2925

Reviewer #1 (Public review): https://doi.org/10.7554/eLife.105005.3.sa1
Reviewer #2 (Public review): https://doi.org/10.7554/eLife.105005.3.sa2
Reviewer #3 (Public review): https://doi.org/10.7554/eLife.105005.3.sa3
Author response https://doi.org/10.7554/eLife.105005.3.sa4

## Additional files

### Supplementary files

Supplementary file 1. Features, descriptors and protocol summary. (**a**) Hydrophobic nodes in SHV-1. (**b**) Dynamic descriptors and number of residue pairs detected. (**c**) Number of residues in each community detected using only the coevolution scores. (**d**) Summary of molecular dynamics simulation systems.

MDAR checklist

## Data availability

All files required to run the simulations (topology, coordinates, input), processed trajectories (xtc), corresponding coordinates (pdb), can be downloaded from https://doi.org/10.57760/sciencedb.15876 (PDC-3) and https://doi.org/10.5281/zenodo.13693144 (SHV-1). DyNoPy is available at https://github.com/alepandini/DyNoPy, (copy archived at *Pandini, 2024*).

The following previously published datasets were used:

| Author(s) | Year | Dataset title | Dataset URL | Database and Identifier |
|-----------|------|---------------|-------------|-------------------------|
| Haider S | 2024 | Ω-Loop mutations control the dynamics of the active site by modulating a network of hydrogen bonds in PDC-3 β-lactamase | https://doi.org/10.57760/sciencedb.15876 | Science Data Bank, 10.57760/sciencedb.15876 |
| Haider S | 2024 | Functionally Important Residues from Graph Analysis of Co-evolved Dynamical Couplings | https://doi.org/10.5281/zenodo.13693144 | Zenodo, 10.5281/zenodo.13693144 |

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

# Appendix 1

## Interaction energies (E)

The pairwise non-bonded interaction energies for the upper triangle of the N×N residue matrix are calculated using dyno_pwie.py, which is a wrapper to ccptraj from ambertools. The list of pairs is generated and depending on the number of available threads ($n_t$), '$n$' sets of pairs are created, and each pair set is assigned to an instance of ccptraj for calculation of pairwise interaction energies. As this is a RAM-dependent operation, based on the size of the MD trajectory, care must be taken to not spawn too many instances of ccptraj as it will slow down each instance of the ccptraj calculation. For each residue pair ($r_{ij}$), a separate file with coulomb ($q_{ij}$) and van der Waals ($w_{ij}$) interaction energies, as two separate columns, are saved to a h5py compressed file and the total interaction energy ($\epsilon_{ij}$ for each time step can be calculated from this data using *Equation A1*.

$$\epsilon_{ij} = q_{ij} + w_{ij} \tag{A1}$$

## Collective variables description

### Radius of gyration (Rg)

The radius of gyration (Rg) measures the compactness of the protein structure through the calculation of the root mean square distance of all atoms from the centre of mass. A lower Rg indicates a more compact structure, while a higher Rg suggests a more expanded or unfolded state. This CV was calculated using the MDAnalysis v2.7.0 (*Gowers et al., 2016*; *Michaud-Agrawal et al., 2011*), focusing on the whole protein for both enzymes.

### The first principal component (PC1)

Principal component analysis identifies the major dynamical feature of a protein by reducing the dimensionality directly from molecular dynamics simulation data. PC1 represents the largest variance on the selected features across the simulation. Global PC1 was computed using PyEMMA v2.5.12 (*Scherer et al., 2015*), focusing on backbone torsion angles and $\chi 1$ angles for all residues, capturing the global dominant structural changes.

### The partial PC1 (PC1_partial)

PC1_partial was calculated using the same methodology as PC1 but only focusing on functionally significant residues and essential secondary structures. This approach prioritizes the dynamics within regions critical to the protein's function, thereby providing a more targeted analysis of pertinent conformational changes. It is particularly well-suited for studying proteins that are predominantly rigid with localized flexible regions.

### Time-lagged independent component 1 (TC1)

Time-lagged independent component analysisidentifies slow, independent processes within the protein dynamics. TC1 captures the slowest conformational changes in the protein over time. It was computed globally using backbone torsion angles and $\chi 1$ angles for all residues, with PyEMMA v2.5.12 (*Scherer et al., 2015*).

### Partial time-lagged independent component 1 (TC1_partial)

Similarly, TC1_partial was calculated, but focused on specific, functionally important regions. By analysing these key residues and structures, TC1_partial represents the slow conformational changes crucial within essential regions, which are highly corresponding with protein functions.

### Global root mean square deviation (gRMSD)

The RMSD measures the average deviation of a protein's atomic positions from a reference structure, typically the starting conformation. gRMSD provides an overview of the protein's structural deviation over time. In our case, trajectories were first aligned, and Cα RMSD was calculated using MDAnalysis v2.7.0 (*Gowers et al., 2016*; *Michaud-Agrawal et al., 2011*), utilizing the starting conformation of each protein as a reference.

### Partial root mean square deviation (pRMSD)

This CV is calculated similarly to gRMSD but focuses on specific residues or regions of interest. pRMSD provides insight into the structural stability of functionally important areas of the protein by focusing on catalytical residues and essential loops.

### Dynamical root mean square deviation (dRMSD)

dRMSD is an extension of RMSD that considers the dynamic nature of protein motions by analysing fluctuations over time rather than static deviations. More specifically, instead of calculating RMSD to a fixed reference structure, it focuses on the difference between adjacent frames. This approach allows the analysis of short-timescale fluctuations and provides insight into the dynamic nature of protein motions over time. It was also calculated using MDAnalysis v2.7.0 (*Gowers et al., 2016*; *Michaud-Agrawal et al., 2011*) focusing on the Cα and offers a more nuanced understanding of the protein's conformational landscape.

### Global solvent-accessible surface area (gSASA)

SASA quantifies the surface area of the protein accessible to the solvent. gSASA was calculated using GROMACS v2020.1 (*Abraham et al., 2015*) and provides information about the protein's overall exposure to the solvent, which is relevant for understanding folding and binding interactions.

### Partial solvent-accessible surface area (pSASA)

pSASA focuses on the solvent exposure of specific active site residues, offering insights into how binding sites or functional regions of the protein interact with the solvent.

### Active site pocket volume

The volume of the active site pocket was calculated using Mdpocket (*Schmidtke et al., 2011*). This CV is important for understanding the size and shape changes in the binding site, which can influence ligand binding and protein function. Upon importing the topology file and trajectories, Mdpocket autonomously identifies potential binding pockets within the protein structure. Subsequently, the relevant pocket was manually selected (*Figure 2—figure supplement 3*)

### Number of hydrogen bonds (hbond)

The number of hydrogen bonds was calculated per frame using VMD v1.9.3 (*Humphrey et al., 1996*), with a distance threshold of 3.5 Å and an angle cut-off of 40°. This CV provides insight into the stability of the protein structure and interactions that are critical for maintaining its conformation and function.

### Residues and regions used in partial CVs

For SHV-1, the backbone torsion angles and $\chi 1$ angles for catalytic important residues—S70, T71, K73, S130, N132, K234, and T235—and Ω-loop residues (R164-D179) were used as the input feature to calculate the partial CVs. Similarly, for PDC-3, the backbone torsion angles and $\chi 1$ angles for catalytic important residues—K67, Y150, N152, K315, T316, and G317—Ω-loop residues (G183-S226), and R2-loop residues (L280-Q310) were utilized to calculate all the partial variables.

# Appendix 2

## Other SHV-1 communities

Core residues for some of the communities identified by DyNoPy have never been studied in class A β-lactamases before. DyNoPy predicts these residues as essential due to their coevolution trends with many other residues and their critical role in class A β-lactamase dynamics (*Appendix 2—figure 1*).

D$_{267}$ is the core residue in community 3 (*Appendix 2—figure 1A*). This residue is located on the loop connecting β-sheet β9 and α-helix α12. It is outside the catalytic site and has not undergone essential substitutions, thus remaining unexplored in class A β-lactamases. However, DyNoPy indicates its potential importance due to its relationships with three known essential mutation sites: L$_{35}$, E$_{240}$, and A$_{187}$. L$_{35}$ and E$_{240}$ are predominant ESBL mutation sites, while A$_{187}$ substitution confers an inhibitor-resistant phenotype (*Chang et al., 2001*). The relationship between D$_{267}$ and these mutation sites suggests a trend towards coevolution, and interactions between D$_{267}$ and these mutation sites are important for protein dynamics.

Similarly, I$_{279}$, located also on α12, is the core residue of community 8 (*Appendix 2—figure 1B*). This residue forms relationships with R$_{244}$, a critical inhibitor resistant mutation site (*Giakkoupi et al., 1998*). Other residues within this community are predominantly positioned on the α1 and α12 helices, near the protein's terminus, highlighting the significant contribution of I$_{279}$ to protein integrity and stability.

For community 9, I$_{155}$ serves as the core residue (*Appendix 2—figure 1C*). This community is relatively localized and does not encompass any known essential residues. Most of the residues involved in this community are spatially close to each other, suggesting that I$_{155}$ plays a vital role in the local dynamics, especially surrounding α-helices α7 and α9.

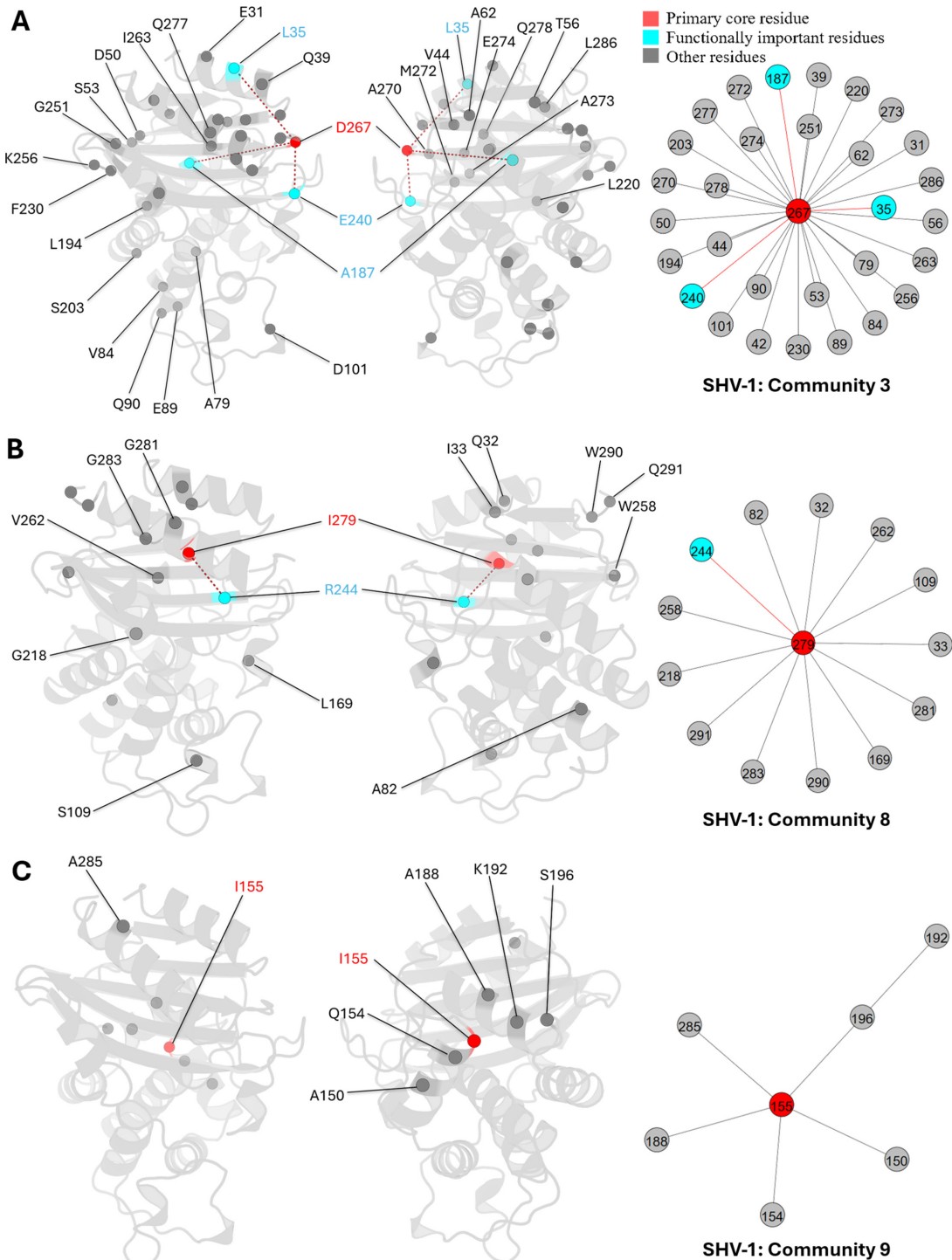

**Appendix 2—figure 1.** Communities 3, 8, and 9 of SHV-1 β-lactamase. All the residues are depicted as spheres on the protein structure. The core residue for each community is D267, I279, and I155, respectively. They are highlighted in red. Functional important residues are marked in cyan. (**A**) Community 3 of SHV-1, comprising 31 residues. (**B**) Community 8 of SHV-1, containing 14 residues. (**C**) Community 9 of SHV-1, with 7 residues.

## Other PDC-3 communities

The core residues in the other PDC-3 communities have not been extensively studied. However, their detection by DyNoPy suggests a significant trend in co-evolution and highlights their crucial role in protein dynamics (***Appendix 2—figure 2***). This emphasizes the capability of DyNoPy to predict

essential residues in previously unexplored proteins, potentially offering valuable insights for future experimental research.

$E_{49}$, $D_{206}$, and $R_{210}$ are core residues for community 3, a small community containing only 14 residues (*Appendix 2—figure 2A*). $R_{210}$ is the primary core residue in this community, linking to six residues, while both $E_{49}$ and $R_{210}$ are secondary core residues that show a relationship with four residues. Unlike other communities with widespread interactions, community 3 illustrates a localized relationship among Ω-loop residues, primarily on the short β5a and β5b β-sheets and the short helix α7a. Also, 8 out of 14 residues are Ω-loop residues, with the remainder located on adjacent loops or near the terminals of adjoining secondary structures, all of which are flexible regions. This community indicates that $R_{210}$ is crucial for maintaining local structural integrity and the stability of the Ω-loop.

$G_{202}$, the core residue of community 4, appears essential for active site stability by interacting with residues whose side chains point into the active site (*Appendix 2—figure 2B*). It also stabilizes two α-helices, α2 and α5, located adjacent to the active site. $P_{154}$ is a special mutation site in the PDC family. $P_{154}$L occur in PDC-73 and PDC-81, giving the protein a mild increase in resistance to ceftazidime (*Berrazeg et al., 2015*).

$K_{204a}$ and $R_{207}$ are the central residues in community 5, each establishing six interactions and thus sharing an equal position of importance within this community. $K_{281}$ and $K_{351}$, which interact with both core residues, are highlighted in light yellow (*Appendix 2—figure 2C*). $K_{204a}$ forms direct interactions with six residues, primarily located on the opposite side of the active site, either on the R2 loop or at the beginning of helix α11. This suggests that $K_{204a}$ plays a crucial role in maintaining the conformation of the R2 site. Additionally, $K_{204a}$ shows direct correlations with the catalytically significant residue $K_{315}$. Although $R_{207}$ is spatially close to $K_{204a}$, it interacts with residues on loops that are distant from the catalytic site.

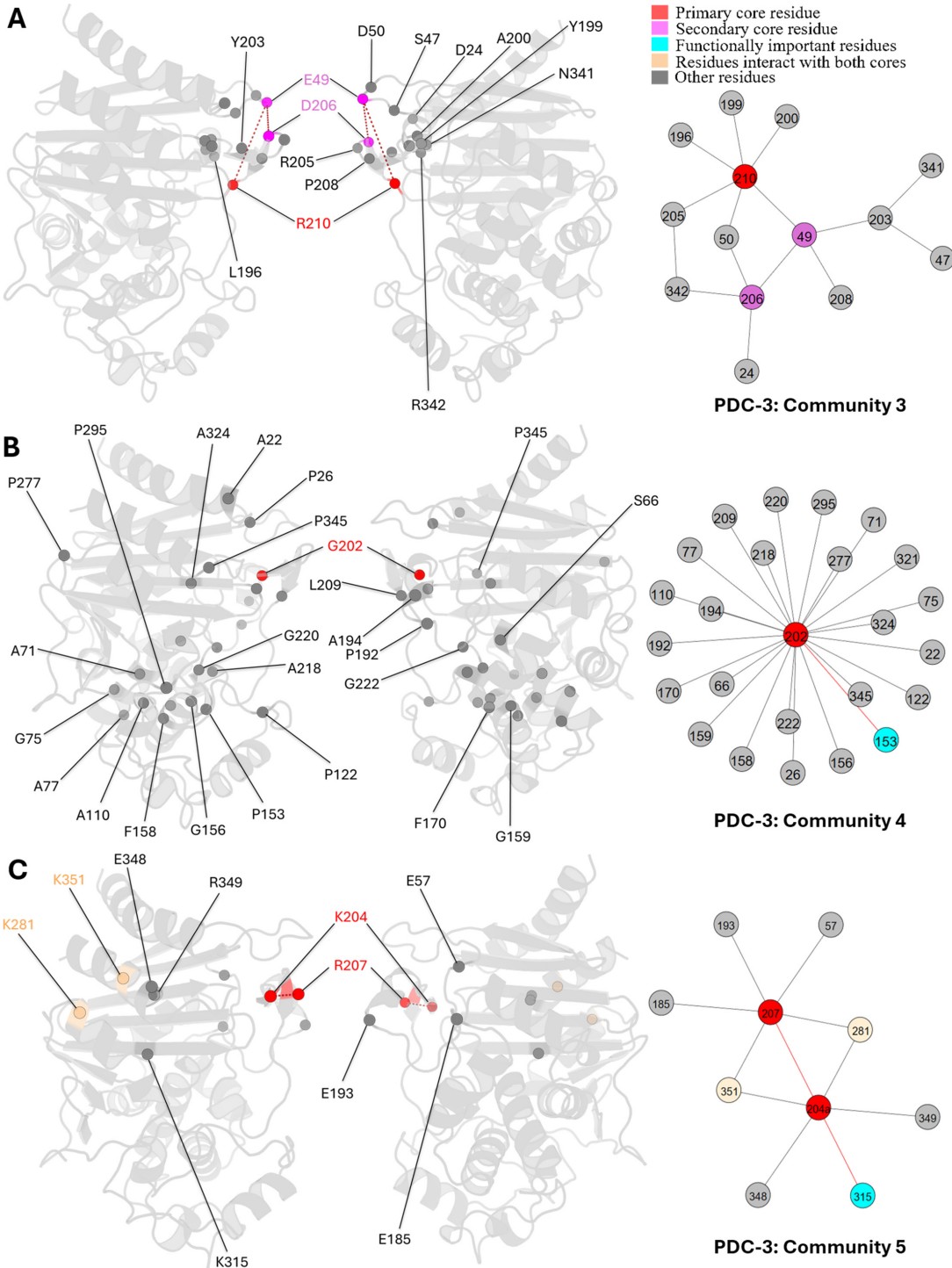

**Appendix 2—figure 2.** Communities 3, 4, and 5 of PDC-3 β-lactamase. All the residues are depicted as spheres on the protein structure. The core residue for each community is highlighted in red, while purple is used to emphasize the secondary core residue. Residues that interact with both cores are coloured in light yellow. Functional important residues are marked in cyan. (**A**) Community 3 of PDC-3, comprising 14 residues with $R_{210}$ being the primary core residue. (**B**) Community 4 of PDC-3, containing 25 residues and is centred by $G_{202}$. (**C**) Community 5 of PDC-3, embracing 10 residues and having two core residues $K_{204}$ and $R_{207}$.

