## [Editor Report · eLife Assessment]

This article reports the analysis of coevolutionary patterns and dynamical information for identifying functionally relevant sites. These findings are considered **important** due to the broad utility of the unified framework and network analysis capable of revealing communities of key residues that go beyond the residue-pair concept. The data are **solid** and the results are clearly presented.

---

## [Referee Report · Reviewer #1 (Public review)]

Summary:

As reported above, this paper by Xu et al reports on a new method to combine the analysis of coevolutionary patterns with dynamic profiles to identify functionally important residues and reveal correlations between binding sites.

Strengths:

In general, coevolutionary analysis and MD analysis are carried out separately and while there have been attempts to compare the information provided by the two, no unified framework exists. Here, the authors convincingly demonstrate that integrating signals from Dynamics and coevolution gives information that substantially overcomes the one provided by either method in isolation. While other methods are useful, they do not capture how dynamics is fundamental to define function and thus sculpts coevolution, via the 3D structure of the protein. At the same time, the authors demonstrate how coevolution in turn also influences internal dynamics. The Networks they rebuild unveil information at an even higher level: the model starts pairwise but through network representation the authors arrive to community analysis, reporting on interaction patterns that are larger than simple couples.

Comments on latest version:

I have nothing to add to this revision. The paper looks excellent and very interesting.

---

## [Referee Report · Reviewer #2 (Public review)]

Summary:

The authors introduced a computational framework, DyNoPy, that integrates residue coevolution analysis with molecular dynamics (MD) simulations to identify functionally important residues in proteins. DyNoPy identifies key residues and residue-residue coupling to generate an interaction graph and attempts to validate using two clinically relevant β-lactamases (SHV-1 and PDC-3).

Strengths:

DyNoPy could not only show clinically relevance of mutations but also predict new potential evolutionary mutations. Authors have provided biologically relevant insights into protein dynamics which can have potential applications in drug discovery and understanding molecular evolution.

Comments on latest version:

I appreciate the efforts of the authors to address my comments.

---

## [Referee Report · Reviewer #3 (Public review)]

Summary:

In this paper, Xu, Dantu and coworkers report a protocol for analyzing coevolutionary and dynamical information to identify a subset of communities that capture functionally relevant sites in beta-lactamases.

Strengths:

The combination of coevolutionary information and metrics from MD simulations is interesting for capturing functionally relevant sites, which can have implications in the fields of drug discovery but also in protein design.

Comments on latest version:

The authors have successfully addressed all my previous comments/concerns. I am happy with the current version of the manuscript.

---

## [Author Response]

The following is the authors’ response to the original reviews.

**Reviewer #1 (Public review):**
Summary:As reported above, this paper by Xu et al reports on a new method to combine the analysis of coevolutionary patterns with dynamic profiles to identify functionally important residues and reveal correlations between binding sites.Strengths:In general, coevolutionary analysis and MD analysis are carried out separately and while there have been attempts to compare the information provided by the two, no unified framework exists. Here, the authors convincingly demonstrate that integrating signals from Dynamics and coevolution gives information that substantially overcomes the one provided by either method in isolation. While other methods are useful, they do not capture how dynamics is fundamental to define function and thus sculpts coevolution, via the 3D structure of the protein. At the same time, the authors demonstrate how coevolution in turn also influences internal dynamics. The Networks they rebuild unveil information at an even higher level: the model starts pairwise but through network representation the authors arrive to community analysis, reporting on interaction patterns that are larger than simple couples.Weaknesses:The authors should- Make an effort in suggesting/commenting the limits of applicability of their method;

We have added a sentence on Page 17, line 15 that describes the limitation of our method.

- Expand discussion on how DyNoPy compares to other methods;

A paragraph has been added to explain the comparison with other models (Page 3, line 18)

- Dynamic is not essential in all systems (structural proteins): The authors may want to comment on possible strategies they would use for other systems where their framework may not be suitable/applicable.

We agree with the reviewer that dynamics is not essential in all systems. In systems where there is limited role of dynamics in the function, the analysis done with DyNoPy is equivalent to conventional coevolution analysis, which can be consider one limitation of our method. Conversely, for dynamic proteins, combining functional dynamics descriptors with coevolution analysis using DyNoPy, helps in denoising information by deconvolution of communities. We have included this in the manuscript to highlight the suitability/applicability of the method.

Further, we have added a paragraph in the Introduction and conclusions highlighting the main difference between DyNoPy and existing computational tools like DCCM, KIN, and SPM and for your convenience it is provided below:

“Functional sites are often regulated by both, local and global interactions. Changes in these interactions are instrumental for functional events like substrate binding, catalysis, and conformational changes (18). The development of physical models of protein dynamics and the increase in available computational power has stimulated the adoption of computational techniques (19, 20) to investigate the conformational dynamics of proteins, an essential component of the many biological functions (21, 22). Different models have been proposed to describe the interactions between residues during simulations and network models have been particularly popular, including methods on single structures and MD simulations data built by analysing the response to external forces on residue networks (23), by estimating the prevalence of non-covalent energy interaction networks in homologous proteins (24), or by analysing linear or non-linear correlation in atomic fluctuations (25, 26). These techniques have demonstrated their usefulness in extracting allosteric networks from structural data with applications in enzyme design (26).”

**Reviewer #2 (Public review):**
Summary:Authors introduced a computational framework, DyNoPy, that integrates residue coevolution analysis with molecular dynamics (MD) simulations to identify functionally important residues in proteins. DyNoPy identifies key residues and residue-residue coupling to generate an interaction graph and attempts to validate using two clinically relevant β-lactamases (SHV-1 and PDC-3).Strengths:DyNoPy could not only show clinically relevance of mutations but also predict new potential evolutionary mutations. Authors have provided biologically relevant insights into protein dynamics which can have potential applications in drug discovery and understanding molecular evolution.Weaknesses:Although DyNoPy could show the relevance of key residues in active and non-active site residues, no experiments have been performed to validate their predictions.

We thank the reviewer for highlighting this point. We acknowledge that direct experimental validation of our predictions for DyNoPy has not yet been performed. However, we have provided explanations and evidence from experiments conducted on closely related homologs to support the relevance of key residues. These homologs share significant structural and functional similarity, which strengthens the reliability of our predictions.

In addition, they should compare their method with conventional techniques and show how their method could be different.

We thank all the reviewers for highlighting this oversight on our behalf. In Introduction and conclusion, we have added the following paragraphs:

“Functional sites are often regulated by both, local and global interactions. Changes in these interactions are instrumental for functional events like substrate binding, catalysis, and conformational changes (18). The development of physical models of protein dynamics and the increase in available computational power has stimulated the adoption of computational techniques (19, 20) to investigate the conformational dynamics of proteins, an essential component of the many biological functions (21, 22). Different models have been proposed to describe the interactions between residues during simulations and network models have been particularly popular, including methods on single structures and MD simulations data built by analysing the response to external forces on residue networks (23), by estimating the prevalence of non-covalent energy interaction networks in homologous proteins (24), or by analysing linear or non-linear correlation in atomic fluctuations (25, 26). These techniques have demonstrated their usefulness in extracting allosteric networks from structural data with applications in enzyme design (26). ”

An explanation of "communities" divided in the work and how these communities are relevant to the article should be provided. In addition, choice of collective variables and their relevance in residue coupling movement is also not very well explained. Dynamics cross correlation map can also be a good method for understanding the residue movements and can explain the residue-residue coupling, it is not explained how DyNoPy is different from the conventional methods or can perform better.

The following sentences have been included in the manuscript to address the questions raised by the reviewer:

On Community Definition and relevance

DyNoPy identified coevolving residue pairs (scaled coevolution score >1) with interactions strongly correlated with protein functional motions (i.e., J values larger than zero). Applying network analysis on the combined dynamics-coevolution matrix helps us extracting higher-order interactions beyond pairwise coupling and detecting critical residues, which show multiple interactions with each other. Moreover, indirect long-range relationships, which would be hard to identify from numerical data, could be detected through community clustering. Community-based analysis offers a more comprehensive understanding of residue relationships and enables the visualization of residue couplings on the protein structure.

On Choice of collective variables:

DyNoPy works on the assumption that time-dependent interactions between critical residues, either having significant structural change or not will correlate with functional conformational motions. Since MD simulation data is high-dimensional, a time-dependent dynamic descriptor is required to extract the most relevant information for the process under study. A good collective variable (CV) should appropriately describe protein functional motions. Thus, a CV that detects the highest number of residue couplings is expected to be the most suitable descriptor (Mentioned in Page 22 Line 14). In our study, we tested 12 CVs, either focusing on the entire protein or on selected regions. And the best performed CV (the one identified the most residue couplings) was selected for further analysis. In practical applications, users can decide whether to focus on the most relevant global or local dynamics descriptor depending on the dynamics of their specific system.

We have added a paragraph in the Introduction differentiating DyNoPy with other methods including DCCM. DCCM differs from DyNoPy in two aspects (1) it does not account for inter-residue coevolution (2) the correlation matrix captures correlations of atomic/residue movements associated with the whole intrinsic dynamics of the system, without filtering for the contributions to the important motions involved in the biological function. Additionally, any residue pair contributing to functional motion without itself undergoing any structural change will not be visible in this approach.

In the sentence "DyNoPy identified eight significant communities of strongly coupled residues within SHV-1 (Supporting Fig. S4A)" I could not find a clear description of eight significant communities.

The following sentences have been included in the results, methods and figure legends that define ‘significant community’:

‘DyNoPy identified eight meaningful communities, each consisting of at least three strongly coupled residues within SHV-1 (Supplementary Fig. S4A). All crucial catalytic residues and critical substitution sites previously mentioned participating in one of these communities with the exceptions of R_43_, R_202_, and S_130_.’ (Page 8 Line 28)

‘A meaningful community should contain at least three residues.’ (Page 21 Line 2)

‘A reasonable residue community should contain at least three residues.’ (SI Page 11)

Again the description of communities is not clear to me in the following sentence "Detailed description of the other three communities is provided in the supporting information (Fig. S6)."

This following sentence has been rewritten.

‘Detailed description of communities with secondary importance for protein function (community 3, 8, and 9) is provided in the supplementary information (Supplementary Fig. S6).’ (Page 9, line 8)

In the sentence "N170 acts as an intermediary between N136 and E166". Kindly cite the reference figure to show N179 as intermediate residue.

This sentence has been rewritten to avoid any confusion.

‘Although DyNoPy did not detect this direct interaction between N136 and E166, the established relationship between N136 and N170 highlights the role of N136 in influencing E166.’ (Page 10 Line 8)

Please be careful with the numbers. In the sentence "These residues not only interact with each other directly but are also indirectly coupled via 21 other residues." I could count 22 other residues and not 21.

We thank the reviewer for spotting this error. This has now been corrected. All the communities are counted again.

‘These residues not only interact with each other directly but are also indirectly coupled via 22 other residues.’ (Page 12 Line 14)

In the sentence "Unlike other substitution sites that are adjacent to the active site, R_205_ is situated more than 16 Å away from catalytic serine S_70_". Please add this label somewhere in the figure.

The figure legends have been updated to include this. Distances have been added to community 4 Fig. 3 and community 6 Fig. 4. Residue index in the legend of Fig.3 has been included as subscript. Distance in the main text has been changed to be more accurate.

‘G_156_ and A_146_ are two functional important residues distant from the active site. G_156_ is 21.3Å away from the catalytic S_70_. A_146_ is 16.8Å away from S_70_.’ (Page 12 Line 2)

‘R_205_ is a functional important residue that is 20.6Å away from the active site S_70_.’ (Page 13 Line 10)

Please cite a reference in the sentence "This indicates that mutations on G238 would result in an alteration on protein catalytic function, as well as an increased flexibility of the protein, which strongly aligns with previous finding."

The citation has been added

‘This indicates that mutations on G238 would result in an alteration on protein catalytic function, as well as an increased flexibility of the protein, which strongly aligns with previous finding (62).’ (Page 15 Line 2)

**Reviewer #3 (Public review):**
Summary:In this paper, Xu, Dantu and coworkers report a protocol for analyzing coevolutionary and dynamical information to identify a subset of communities that capture functionally relevant sites in beta-lactamases.Strengths:The combination of coevolutionary information and metrics from MD simulations is interesting for capturing functionally relevant sites, which can have implications in the fields of drug discovery but also in protein design.Weaknesses:The combination of coevolutionary information and metrics from MD simulations is not new as other protocols have been proposed along the years (the current version of the paper neglects some of them, see below), and there are a few parameters of the protocol that, in my opinion, should be better analyzed and discussed.(1) As mentioned, the introduction of the paper lacks some important publications in the field of using graph theory to represent important interaction networks extracted from MD simulations (DOI: 10.1002/pro.4911), and also combining MD data with MSA to identify functionally relevant sites for enzyme design (doi: 10.1021/acscatal.4c04587, 10.1093/protein/gzae005).

We are very grateful for pointing us to these references. We have added a paragraph in the Introduction mentioning these and other computational tools similar to DyNoPy. Further, in conclusion we have highlighted the differences between DyNoPy and existing tools.

(2) The matrix used to apply graph theory (J_ij) is built from summing the scaled coevolution and degree of correlation values. The alpha and beta weights are defined, and the authors mention that alpha is set to 0.5, thus beta as well to fulfil with the alpha + beta = 1. Why a value of 0.5 has been selected? How this affects the overall results and conclusions extracted? The finding that many catalytically relevant residues are identified in the communities is not surprising given that such sites usually present a high conservation score.

This is an excellent question. Our present formulation allows the user to easily assess the influence of coevolution and dynamic couplings on the output. Setting alpha to 0.5, weights both evolutionary and dynamics information equally and has shown promising results in SHV-1 and PDC-3. As it has been presented in the manuscript, setting alpha to 1, i.e., purely utilising coevolution data does not let us identify critical residues effectively as all residues are included in the set (Supplementary Fig. S4 and S5). In future work, we would like to investigate the effect of scanning alpha from 0 to 1 on the final residue list, possibly on a larger set of proteins and protein families.

We would also like to point out that some of the residue pairs with coevolution scores in the top 1% have J-scores set to 0, as they lacked significant coupling to the functional dynamics.

(3) Another important point that needs further explanation is the selection of the relevant descriptor of protein dynamics. In this study two different strategies have been used (one more global the other more local), but more details should be provided regarding their choice. What is the best strategy according to the authors? Why not using the same strategy for both related systems? The obtained results using one methodology or the other will have a large impact on the dynamical score. Another related point is: what is the impact of the MD simulation length, how the MSA is generated and number of sequences used for MSA construction?

As in the case of many complex proteins, the flow of information occurs in β-lactamases via structural interactions (https://doi.org/10.7554/eLife.66567). These interactions occur both on a local level, as in the case of binding site residues or residues immediately surrounding the binding site; however, there are interactions far away (>20Å) from the binding site that have the ability to alter function. We have obtained this information from extensive surveys of clinical isolates and experimental data. To account for such interactions, a more global approach has to be taken. To answer the reviewer’s question: each system is unique and there is no one-fixed strategy. In short, the method used should be able to denoise information and the user is advised to fine-tune their findings by corroborating with experimental and clinical information.

The length of MD simulations is also system specific. Some systems effectively sample the functional dynamics within a shorter simulation time, while others take a long timescale MD simulation to converge. The results won’t change as long as the simulation has effectively sampled the functional dynamics associated with biological function.

The MSA is generated by the HH-Suite package as mentioned on Page 19 Line 19. More specifically, the MSA is constructed based on the UniRef30 database, where sequences are clustered, and each cluster contains sequences with at least 30% sequence identity. This provides a non-redundant set of protein sequences. Our package allows the automatic generation of MSAs from the database. For SHV-1, the alignment contains 18,175 protein sequences and for PDC-3, the alignment consists of 27,892 protein sequences. Full details of this protocol are published in Bibik et al. (https://doi.org/10.1093/bioinformatics/btae166). We have revised the methods section to include these details.

Other Minor Alterations

‘Fig. S1 and S2’ has been changed to ‘Supplementary Fig. S1 and S2’ for consistency (Page 6 Line 12)

(1) ‘Figure 5B’ has been changed to ‘Fig. 5B’ for consistency (Page 16 Line 11)

(2) All the ‘Figure’ has been changed to ‘Fig.’ in the SI for consistency

(3) Just as the suggestion, an alteration has been made on the Step 1 of Fig.1.